# How outcomes are measured after spontaneous intracerebral hemorrhage: A systematic scoping review

**Sara Massicotte**[1‡]*, **Ronda Lun**[1‡], **Vignan Yogendrakumar**[1], **Brian Dewar**[1], **Hee Sahng Chung**[1], **Ricarda Konder**[1], **Holly Yim**[1], **Alexandra Davis**[2], **Dean Fergusson**[3], **Michel Shamy**[1], **Dar Dowlatshahi**[1]

1 Department of Medicine (Neurology), University of Ottawa Brain and Mind Research Institute and Ottawa Hospital Research Institute, Ottawa, Ontario, Canada, 2 University of Ottawa, Ottawa, Ontario, Canada, 3 Ottawa Hospital Research Institute, Ottawa Methods Centre, University of Ottawa School of Epidemiology, Public Health and Preventative Medicine, Ottawa, Ontario, Canada

‡ SM and RL are contributed equally to this manuscript as co-first authors.
* Smass006@uottawa.ca

## Abstract

**Data Availability Statement:** All relevant data are within the manuscript and its Supporting Information files.

### Background and purpose

Recovery after intracerebral haemorrhage (ICH) is often slower than ischemic stroke. Despite this, ICH research often quantifies recovery using the same outcome measures obtained at the same timepoints as ischemic stroke. The primary objective of this scoping review is to map the existing literature to determine when and how outcomes are being measured in prospective studies of recovery after ICH.

### Methods

We searched MEDLINE, Embase, Cochrane Central Register of Controlled Trials and Web of Science from inception to November 2019, for prospective studies that included patients with ICH. Two investigators independently screened the studies and extracted data around timing and type of outcome assessment.

### Results

Among the 9761 manuscripts reviewed, 395 met inclusion criteria, of which 276 were observational studies and 129 were interventional studies that enrolled 66274 patients. Mortality was assessed in 93% of studies. Functional outcomes were assessed in 85% of studies. The most frequently used functional assessment tool was the modified Rankin Scale (mRS) (60%), followed by the National Institute of Health Stroke Severity Scale (22%) and Barthel Index (21%). The most frequent timepoint at which mortality was assessed was 90 days (41%), followed by 180 days (18%) and 365 days (12%), with 2% beyond 1 year. The most frequent timepoint used for assessing mRS was 90 days (62%), followed by 180 days (21%) and 365 days (17%).

**Funding:** This study did not receive any specific funding or sponsor supports from any funding agency in the public, commercial or not-for-profit sectors. Dr. Dowlatshahi is supported by a Heart and Stroke Foundation of Canada Clinician-Scientist Award and a University of Ottawa Clinical Research Chair.

**Competing interests:** The authors have declared that no competing interests exist.

## Conclusion

While most prospective ICH studies report mortality and functional outcomes only at 90 days, a significant proportion do so at 1 year and beyond. Our results support the feasibility of collecting long-term outcome data to optimally assess recovery in ICH.

## Introduction

Intracerebral hemorrhage (ICH) accounts for 10–20% of all strokes [1], but is associated with a disproportionate degree of mortality, morbidity and economic burden [2]. Historically, ICH has been associated with a 30-day mortality of approximately 40%, with only one in five patients achieving functional independence at six months.

Recovery after ischemic stroke tends to be rapid in the first three months with a subsequent plateau by six months [3]. Conversely, less is known about the long-term natural history and trajectory of recovery in ICH [4, 5] though emerging evidence indicates that ICH recovery may be delayed and prolonged, with survivors continuing to show clinical improvement up to one year [6]. These findings suggest that recovery after ICH should be assessed beyond three months to ensure that patient outcomes are adequately captured and that determinations of treatment efficacy are accurate. Yet it is unclear whether clinical trials and observational studies follow subjects beyond three months.

The primary objective of our scoping review is to describe the timing of outcome assessment undertaken by prospective studies of patients with spontaneous ICH. Our secondary objective is to determine which assessment scales are used to measure outcomes.

## Materials and methods

### Study protocol and registration

The authors declare that all supporting data and methodological detail are available within the article and online-only supplement. The protocol for this study was previously published [7] and registered at the University of Ottawa Research Repository. This study was conducted based on the guidelines of the Johana Briggs Institute (JBI) Methodology for Scoping Reviews [8] and complies with the PRISMA extension statement for scoping reviews [9].

### Eligibility criteria and search strategy

We included prospective observational and interventional studies of adult patients (≥18 years of age) presenting with spontaneous ICH, confirmed with either CT or MRI. Included studies required assessment of a functional outcome, defined as any standardized measurement scale that assessed and quantified patients' clinical status or ability to function in life after their ICH. We excluded studies of patients with a known secondary aetiology and studies of non-parenchymal haemorrhage (eg. subarachnoid, subdural, epidural, isolated intraventricular hemorrhage). Studies focusing solely on recurrence of ICH were excluded. Studies without a standardized timing of outcome assessment, for example at discharge, were also excluded. Studies without planned repeat outcome assessments, such as retrospective studies, case series and case reports, were excluded.

Our search strategy included the following four databases from the date of inception to November 2019: MEDLINE, Embase, Cochrane Central Register of Controlled Trials, and Web of Science. A search strategy was developed (see S1 File), with the assistance of an

information specialist (AD), using search terms specific to the database being searched. We only included studies published in English.

## Study selection

Screening and full-text review was conducted using Covidence Systematic Review software (Covidence, Melbourne, Australia). Two independent reviewers screened articles in a two-step manner (SM, RL). In step one, abstracts and titles were screened for potentially relevant articles. Potentially relevant articles then proceeded to full article screening (Step two), using a standardized form. Disagreements in either step were resolved by consensus.

## Data collection and synthesis of results

Data extraction was conducted independently by multiple reviewers (SM, RL, DC, DK and HY) using an *a priori* collection form. We collected publication information, study population information, and outcome data. Patient demographics included: total number of patients, primary country of recruitment, age, sex and history of hypertension. All functional outcomes reported were recorded including the first and last time point of assessment, and whether there were additional assessments within that time frame. The specified start time of the outcome assessment was recorded (eg. at admission, randomization, event, etc.) as well as the specified primary outcomes and the outcome assessor. These results were described qualitatively. As per scoping review guidelines, a formal assessment of methodological quality was not performed [9].

## Results

### Study selection

Among the 9761 studies retrieved, title and abstract screening narrowed our search to 989 manuscripts, of which 395 met inclusion criteria. Reasons for exclusion are listed in Fig 1. The included studies are listed in S1 Table.

### Study and patient characteristics

Of the 395 included studies, 276 were observational studies and 129 were randomized controlled trials, representing a total of 66,274 patients. The median year of publication was 2013 (range 1989–2019). Participant age was reported in 294/395 (74%) studies, sex was reported in 308/395 (78%) studies, and history of hypertension was reported in 200/395 (51%) studies. Of the studies that reported age, 252 of the ages were reported as a mean, and the aggregate mean of the reported ages is 62.4 years. The remaining 40 studies reported age as a median age ranging from 49.6–74 years, and two studies reported the age as a range. Where reported, 32,540/54,891 (59%) were male and 28,348/38,783 (73%) had a history of hypertension.

### Mortality assessment

Mortality was assessed in 369 of 395 studies (93%) (Fig 2). Mortality was more frequently assessed in observational studies (96%) compared to interventional studies (87%). Overall, the most frequent time point at which mortality was assessed was 90 days, followed by 180 days and 365 days; 45 studies (11%) assessed mortality at 365 days, and only 9 studies (2%) assessed mortality beyond 365 days. The latest assessment of mortality reported in any study was at 4380 days (12 years), in a study that had followed 550 males for 12 years or until death [10]. The studies that assessed mortality at 365 days and beyond were mostly observational (65%).

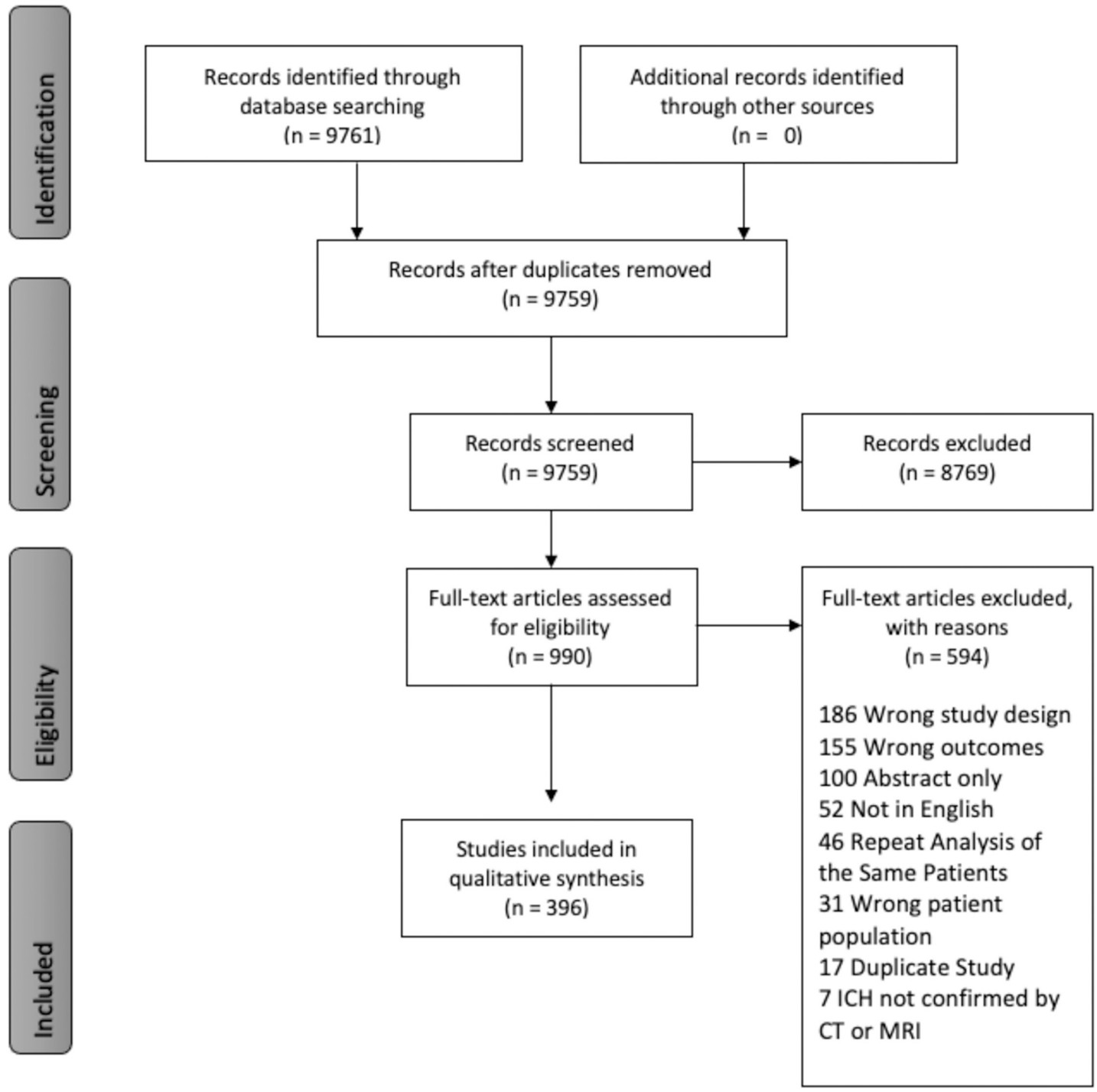

**Fig 1. PRISMA flow diagram of article review process.**

## Functional outcome assessment

Functional outcomes were assessed in 334 of 395 (85%) studies. Seventeen outcome assessment tools were used in at least three studies (Table 1), with 52 outcome tools used in less than three studies (S2 Table). The most frequent outcome assessment tool was the Modified Rankin

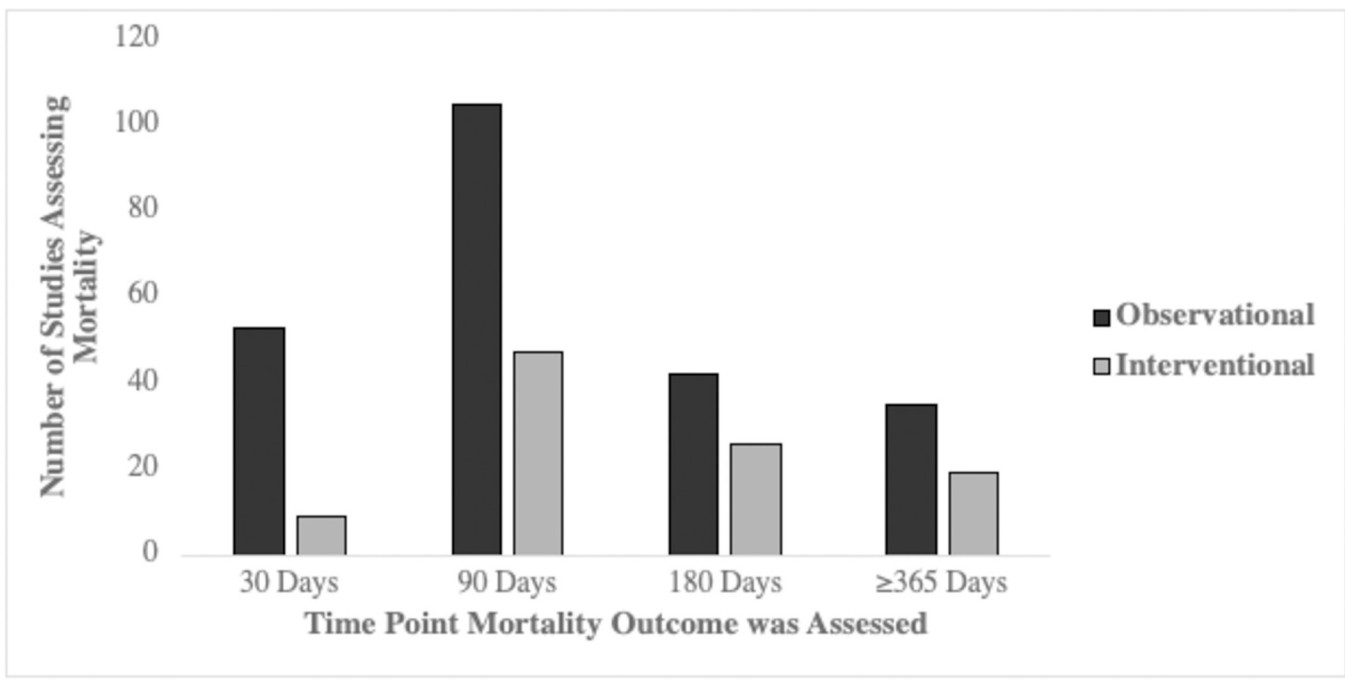

**Fig 2. Mortality assessment frequency in observational and interventional trials.** The number of studies assessing mortality at four time points for observational (black) and interventional (grey) studies respectively.

Scale (mRS), which was reported in 236 studies (60%). The mRS was most frequently assessed at 90 days, but 29 of 236 studies (12%) assessed it at 365 days. The next most common outcome assessment tools, NIHSS and Barthel Index were also most frequently measured at 90 days, with a small subset of studies assessed at 365 days (see Table 1). The studies that assessed functional outcome at 365 days and beyond were mostly observational (63%) and had median date of publication of 2013 (range 1989–2019). Across all outcome assessment methods, 245 of 395 (62%) of the included studies assessed at 90 days, 81 of 395 (21%) at 180 days and 68 of 395 (17%) at or beyond 365 days. Lastly, of the studies that assessed functional outcome at 365 days and beyond, 63% of the studies were funded, 7% were not funded and 30% had undisclosed funding.

## Discussion

We performed a comprehensive systematic scoping review to map the timing of outcome assessment, and the metrics used, to quantify recovery after ICH. We found that the most frequent timepoint at which mortality was assessed was 90 days, but 11% of included studies assessed mortality at one year, and 2% of studies even assessed mortality beyond one year. Similar patterns were seen for functional outcome, with 90 days reported as the most frequent outcome (62%), followed by 180 days (21%) and one year (17%). These results suggest that obtaining long-term outcomes beyond 90 days is feasible in research seeking to measure recovery after ICH.

Our findings are consistent with those from a systematic review conducted in 2000, which reported that 90 days was the most frequent timepoint at which outcomes are measured in studies of post-stroke recovery [11]. However, that study did not distinguish between ischemic stroke and ICH, and only included interventional studies [11]. Our scoping review confirms this is true for ICH, although a significant proportion of studies do assess later outcomes, in

**Table 1. Frequency of outcome measurement tool utilisation and frequency of final time point for each outcome measurement tool utilized in more than three studies.**

| Outcome | Total Number of Uses | Frequency (n = 395 studies) | Last Time Point (days) | Most Frequent Final Time Point in Days (number of studies) | Second Most Frequent Time Point in Days (number of studies) | Third Most Frequent Time Point in Days (number of studies) |
|---|---|---|---|---|---|---|
| Modified Rankin Scale | 236 | 59.75% | 1825 | 90 (137) | 180 (38) | 365 (29) |
| National Institutes of Health Stroke Scale (NIHSS) | 87 | 22.03% | 365 | 90 (18) | 14 (15) | 7 (7)/365 (7) |
| Barthel Index | 81 | 20.51% | 365 | 90 (41) | 180 (14) | 365 (13) |
| Glasgow Outcome Scale | 47 | 11.90% | 365 | 180 (12) | 30 (10) | 90 (7) |
| Glasgow Coma Scale | 37 | 9.37% | 365 | 90 (9) | 14 (5) | 3 (4) / 180 (4) |
| Extended Glasgow Outcome Scale | 11 | 2.78% | 365 | 90 (6) | 180 (2) | 365 (3) |
| Health-Related Quality of Life Scale | 5 | 1.27% | 365 | 365 (2) | 180 (1) | 90 (2) |
| Mini-Mental State Examination (MMSE) | 8 | 2.03% | 365 | 90 (5) | 365 (2) | 90 (5) |
| Modified Barthel Index | 8 | 2.03% | 365 | 180 (4) | 90 (2) | 30 (1)/ 365 (1) |
| European Quality of Life Scale (EuroQol) | 8 | 2.03% | 365 | 90 (6) | 90 (1)/365 (1) | NA |
| Functional Independence Measure | 7 | 1.77% | 365 | 180 (3)/365 (3) | 84 (1) | NA |
| Scandinavian Stroke Scale (SSS) | 6 | 1.52% | 90 | 7 (2)/ 90 (2) | 3 (1)/14 (1) | NA |
| Activities of Daily Living (ADL) Score | 4 | 1.01% | 180 | 180 (2) | 17 (1)/ 90 (1) | NA |
| Stroke impact scale | 4 | 1.01% | 365 | 365 (4) | NA | NA |
| Fugl-Meyer | 3 | 0.76% | 180 | 56 (2) | 180 (1) | NA |
| Modified Telephone Interview for Cognitive Status | 3 | 0.76% | 90 | 90 (3) | NA | NA |
| Health-Related Quality of Life Scale with Quality of Life in Neurological Disorders (NEURO-QOL) | 3 | 0.76% | 365 | 365 (3) | NA | NA |

line with recent evidence suggesting significant differences in recovery trajectories between stroke subtypes [12]. In a large stroke registry-based study from China, the trajectories of recovery were significantly different by stroke subtype; while the percentage of patients with a good outcome plateaued around 6–7 months in ischemic strokes, in hemorrhagic stroke, the plateau in recovery trajectory happened around one year after the event [12]. The reasons for delayed recovery in ICH are unclear, and may relate to hematoma expansion resulting in neurologic deterioration, mass effect, the time required for hematoma resorption, complications from surgical interventions, and the limited availability of treatment options [13, 14]. Measuring outcomes at 90 days in ICH may be too early to detect the full benefits of investigational or standard therapies. For example, the Deferoxamine in Intracerebral Hemorrhage trial (iDEF) [15] showed little evidence of treatment effect at 90 days, but revealed a compelling trend towards efficacy at 6 months. Our review suggests that measuring outcomes beyond 90 days is feasible for both observational and randomized studies, and we suggest future ICH treatment trials incorporate 6 month and one year assessments in their study designs.

A previous review of contemporary literature has found that the mRS and Barthel Index are the most commonly used functional outcome measures in acute stroke trials–which includes ischemic and haemorrhagic stroke [16]. Our results support the finding that the mRS and Barthel Index are also the most commonly used tools for measuring outcomes after ICH, especially since ICH-specific tools have not been created or validated in routine clinical care. We would support the use of these measurement scales given their extensive validation in the literature and ease of education/ implementation across medical professions. Our literature search also found that the NIHSS is another commonly used outcome assessment tool in ICH. While

the NIHSS was initially developed to assess initial stroke severity to determine treatment options during the hyperacute period, it is increasingly being used as an outcome measurement tool in recent stroke literature [17]. Given the breadth of our review, there was clinical heterogeneity across the different outcome scales and their original intended purposes. However, this heterogeneity could not be quantified statistically due to the descriptive nature of this study.

Our study is descriptive in nature and has important limitations. There was variation across studies in terms of the initial reference timepoint; some measured outcomes after study enrolment or randomization, whereas others used the admission date or time of ICH onset. We believe this has little bearing on our conclusions as the resulting difference was small (hours to a few days) relative to the long periods of outcome assessment with which we were interested. We were also unable to verify the method of outcome assessment in each individual study (eg. face-to-face, interview). Furthermore, we excluded any studies with unspecified times for outcome measurements such as "at discharge", as length of stay can vary significantly between patients and jurisdictions. While this approach may underestimate the number of prospective ICH studies, it provides a more precise assessment of feasible timelines. Moreover, our study is strengthened by following an a priori published protocol with a comprehensive systematic search strategy, and by not limiting study type or publication year.

## Conclusion

In this study, we demonstrate that while most prospective ICH research reports both mortality and functional outcomes at 90 days, a significant proportion of both observational and interventional studies do so at one year and beyond. Our findings support the feasibility of collecting longer term outcome data to optimally assess recovery in ICH research. We suggest future ICH studies consider adding 6-month and one-year outcomes whenever possible in order to best capture treatment effect through the full course of recovery.

## Supporting information

**S1 Checklist. PRISMA checklist.**
(DOCX)

**S1 Table. Included studies.**
(DOCX)

**S2 Table. Outcomes in 2 studies.**
(DOCX)

**S1 File.**
(DOCX)

## Author Contributions

**Conceptualization:** Sara Massicotte, Ronda Lun, Vignan Yogendrakumar, Brian Dewar, Alexandra Davis, Dean Fergusson, Michel Shamy, Dar Dowlatshahi.

**Data curation:** Sara Massicotte, Ronda Lun, Hee Sahng Chung, Ricarda Konder, Holly Yim.

**Formal analysis:** Sara Massicotte, Ronda Lun.

**Investigation:** Sara Massicotte, Ronda Lun.

**Methodology:** Sara Massicotte, Ronda Lun, Vignan Yogendrakumar, Brian Dewar, Alexandra Davis, Dean Fergusson, Michel Shamy, Dar Dowlatshahi.

**Project administration:** Ronda Lun.

**Supervision:** Dar Dowlatshahi.

**Validation:** Sara Massicotte, Ronda Lun.

**Visualization:** Sara Massicotte, Ronda Lun.

**Writing – original draft:** Sara Massicotte, Ronda Lun.

**Writing – review & editing:** Sara Massicotte, Ronda Lun, Vignan Yogendrakumar, Brian Dewar, Hee Sahng Chung, Ricarda Konder, Holly Yim, Alexandra Davis, Dean Fergusson, Michel Shamy, Dar Dowlatshahi.

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
