## [Decision Letter · Decision Letter 0]

11 May 2021

PONE-D-21-13140

Natural history of recovery after intracerebral hemorrhage: A Systematic scoping review

PLOS ONE

Dear Dr. Massicotte,

Thank you for submitting your manuscript to PLOS ONE. After careful consideration, we feel that it has merit but does not fully meet PLOS ONE’s publication criteria as it currently stands. Therefore, we invite you to submit a revised version of the manuscript that addresses the points raised during the review process.

We look forward to receiving your revised manuscript.

Kind regards,

Aristeidis H. Katsanos, MD, PhD

Academic Editor

PLOS ONE

Journal Requirements:

Reviewers' comments:

Reviewer's Responses to Questions

**Comments to the Author**

1. Is the manuscript technically sound, and do the data support the conclusions?

Reviewer #1: Partly

Reviewer #2: Yes

2. Has the statistical analysis been performed appropriately and rigorously? 

Reviewer #1: N/A

Reviewer #2: Yes

3. Have the authors made all data underlying the findings in their manuscript fully available?

Reviewer #1: No

Reviewer #2: Yes

4. Is the manuscript presented in an intelligible fashion and written in standard English?

Reviewer #1: Yes

Reviewer #2: Yes

5. Review Comments to the Author

Reviewer #1: This is an interesting scoping review with the aim to determine and summarize the outcome measures and the time points used in prospective studies of recovery after ICH.

Although the authors have performed a thorough literature search, references for included studies are not available and the presented results cannot be evaluated and confirmed. I would suggest developing a supplemental table providing the references of the studies included in the analysis.

The statement “Similar patterns were seen for functional outcome, with 90 days reported as the most frequent outcome (51%), followed by 180 days (22%) and one year (13%)” is presented only in the Discussion Section. Authors may consider presenting the relevant findings in the Results section as well.

Reviewer #2: This is a well conducted scoping review led by Massicotte et. al. with a published protocol that addresses an important question regarding the timeline and type of outcomes used to measure functional outcomes in ICH related research. The paper is clearly written and with well specified objectives, inclusion criteria and search methodology. The research findings are novel because it answers a question specifically related to intracerebral hemorrhage in comparison to prior research that included patients with hemorrhagic and ischemic stroke.

General Comments:

1. I would suggest that authors modify the title to reflect the main objective of scoping review more clearly.

2. First paragraph, did authors mean “functional independency”…?

Background/Methods

Sections are well written with clear primary and secondary objectives

I would suggest authors to include in the methodology that they excluded certain studies ( “we excluded any studies with unspecified times for outcome measurements such as

“at discharge”)

I would also suggest to include de definition that authors utilized for Functional Outcome.

Results

Are adequately presented. Would authors be able to inform percentage of funded trials versus non-funded trials that measured outcomes at 365 days and beyond and include in the discussion as well if possible?

Discussion

I would suggest the authors to discuss the findings of mRS, NIHSS and Barthel Index as most common tools used for ICH research, reasons why authors think that happened and if any recommendations for further ICH research. For example, I noticed that some studies used Glasgow coma scale as a measure of functional outcome… ( suggestion to comment about the heterogeneity of outcomes used)

Discussion of the timelines for outcome assessment (primary objective) is well supported.

Limitations

I suggest authors to include that the method of outcome assessment ( face-to-face, interview etc) was not verified in this review.

Minor suggestions

1)Supplemental Table 2

Misspelling for Modified Ashworth Scale

2)I would suggest including references before the period or the comma.

6. PLOS authors have the option to publish the peer review history of their article (what does this mean?). If published, this will include your full peer review and any attached files.

Reviewer #1: No

Reviewer #2: No

---

## [Author Response · Author response to Decision Letter 0]

15 Jun 2021

Response to Reviewers

Reviewer #1

Comment 1: “This is an interesting scoping review with the aim to determine and summarize the outcome measures and the time points used in prospective studies of recovery after ICH.” Although the authors have performed a thorough literature search, references for included studies are not available and the presented results cannot be evaluated and confirmed. I would suggest developing a supplemental table providing the references of the studies included in the analysis.

Response: Thank you. We have added a table of the references from this study as a supplemental table.

Comment 2: “The statement “Similar patterns were seen for functional outcome, with 90 days reported as the most frequent outcome (51%), followed by 180 days (22%) and one year (13%)” is presented only in the Discussion Section. Authors may consider presenting the relevant findings in the Results section as well.”

Response: Thank you for bringing this to our attention. We have added this to paragraph “Functional Outcome Assessment” of our results section. Moreover, as we reviewed this, we discovered a calculation error in one of our tables and have corrected the original numbers (the correction has no bearing on the interpretation of the study). 

Reviewer #2

Comment 1: This is a well conducted scoping review led by Massicotte et. al. with a published protocol that addresses an important question regarding the timeline and type of outcomes used to measure functional outcomes in ICH related research. The paper is clearly written and with well specified objectives, inclusion criteria and search methodology. The research findings are novel because it answers a question specifically related to intracerebral hemorrhage in comparison to prior research that included patients with hemorrhagic and ischemic stroke.

Response: We thank the reviewer for their comments.

Comment 2: I would suggest that authors modify the title to reflect the main objective of scoping review more clearly.

Response: Thank you. In the modified version of our scoping review, the title is: “How Outcomes are Measured After Spontaneous Intracerebral Hemorrhage: A Systematic Scoping Review”

Comment 3: First paragraph, did authors mean “functional independency”…?

Response: This was a grammatical error and we intended to say, “functional independence”. Thank you for catching this. It has been corrected. 

Comment 4: Sections are well written with clear primary and secondary objectives

I would suggest authors to include in the methodology that they excluded certain studies (“we excluded any studies with unspecified times for outcome measurements such as

“at discharge”)

Response: Thank you for this comment. In the ‘Eligibility Criteria and Search Strategy’ section, we have clarified information about the inclusion and exclusion criteria. We stated, “We excluded studies of patients with a known secondary aetiology and studies of non-parenchymal haemorrhage (eg. subarachnoid, subdural, epidural, isolated intraventricular hemorrhage). Studies focusing solely on recurrence of ICH were excluded. Studies without a standardized timing of outcome assessment, for example at discharge, were also excluded.” 

Comment 5: I would also suggest to include the definition that authors utilized for Functional Outcome.

Response: Thank you for your suggestion. We defined “functional outcome” as any standardized measurement scale that assessed and quantified patients’ clinical status or ability to function in life after their ICH. This definition was chosen to be as inclusive as possible, to ensure the broadest scope of our literature review. This definition has been added to our methodology section under the subheading “Eligibility Criteria and Search Strategy” in the revised manuscript. 

Comment 6: Would authors be able to inform percentage of funded trials versus non-funded trials that measured outcomes at 365 days and beyond and include in the discussion as well if possible?

Response: Thank you for your recommendation. We have added the funding status of the studies that assessed outcomes at 365 days and beyond to the last paragraph of the Results section in the revised manuscript. Indeed, the majority of studies that performed functional assessment at 365 days were funded.

Comment 7: I would suggest the authors to discuss the findings of mRS, NIHSS and Barthel Index as most common tools used for ICH research, reasons why authors think that happened and if any recommendations for further ICH research. For example, I noticed that some studies used Glasgow coma scale as a measure of functional outcome… (suggestion to comment about the heterogeneity of outcomes used)

Response: Thank you. We agree that the discussion of the commonality of mRS, Barthel Index and NIHSS in ICH research would add to our discussion. As such, we have included it in the third paragraph of the Discussions section of our updated manuscript. We added the paragraph below: 

“A previous review of contemporary literature has found that the mRS and Barthel Index are the most commonly used functional outcome measures in acute stroke trials – which includes ischemic and haemorrhagic stroke.(16) Our results support the finding that the mRS and Barthel Index are also the most commonly used tools for measuring outcomes after ICH, especially since ICH-specific tools have not been created or validated in routine clinical care. We would support the use of these measurement scales given their extensive validation in the literature and ease of education/ implementation across medical professions. Our literature search also found that the NIHSS is another commonly used outcome assessment tool in ICH. While the NIHSS was initially developed to assess initial stroke severity to determine treatment options during the hyperacute period, it is increasingly being used as an outcome measurement tool in recent stroke literature.(17) Given the breadth of our review, there was clinical heterogeneity across the different outcome scales and their original intended purposes. However, this heterogeneity could not be quantified statistically due to the descriptive nature of this study.”

Comment 8: Discussion of the timelines for outcome assessment (primary objective) is well supported.

Response: Thank you. 

Comment 9: I suggest authors to include that the method of outcome assessment (face-to-face, interview etc) was not verified in this review.

Response: Thank you. In the revised discussion we have added this limitation. 

Comment 10: Supplemental Table 2 - Misspelling for Modified Ashworth Scale

Response: Thank you. This has been corrected in Supplement Table 2 document. 

Comment 11: I would suggest including references before the period or the comma.

Response: Thank you. This has been corrected.

---

## [Editor Report · Decision Letter 1]

17 Jun 2021

How Outcomes are Measured After Spontaneous Intracerebral Hemorrhage: A Systematic Scoping Review

PONE-D-21-13140R1

Dear Dr. Massicotte,

We’re pleased to inform you that your manuscript has been judged scientifically suitable for publication and will be formally accepted for publication once it meets all outstanding technical requirements.

Kind regards,

Aristeidis H. Katsanos, MD, PhD

Academic Editor

PLOS ONE
---

## [Editor Report · Acceptance letter]

21 Jun 2021

PONE-D-21-13140R1 

How Outcomes are Measured After Spontaneous Intracerebral Hemorrhage: A Systematic Scoping Review 

Dear Dr. Massicotte:

I'm pleased to inform you that your manuscript has been deemed suitable for publication in PLOS ONE. Congratulations! Your manuscript is now with our production department. 

Kind regards, 

on behalf of

Dr. Aristeidis H. Katsanos 

Academic Editor

PLOS ONE